# Uncovering the Inherent Size Dependence of Yield Strength and Failure Mechanism in Micron-Sized Metallic Glass

**DOI:** 10.3390/ma15186362

**Published:** 2022-09-13

**Authors:** Yun Teng, Zhen-Dong Sha

**Affiliations:** 1Department of Mechanical Engineering, City University of Hong Kong, Hong Kong 999077, China; 2State Key Laboratory for Strength and Vibration of Mechanical Structures, School of Aerospace Engineering, Xi’an Jiaotong University, Xi’an 710049, China

**Keywords:** metallic glass micropillar, size dependence, yield strength, failure mechanism, shear band

## Abstract

The sample size effect on the deformation behavior of metallic glasses (MGs) has recently become research of intense interest. An inverse sample size effect is observed in previous experimental studies; where the yield strength decreases with decreasing sample size, rather than increasing. We propose a theoretical analysis based on the shear banding process to rationalize the inherent size dependence of yield strength, showing an excellent agreement with experimental results. Our model reveals that the anomalous inverse size effect is, in fact, caused by a transition in failure mode; from a rapid shear banding process with a shear band (SB) traversing the entire sample in bulk MGs, to an immature shear banding process with propagated SBs only at the surface in micron-sized MGs. Our results fill the gap in the current understanding of size effects in the strength and failure mechanism of MGs at different length scales.

## 1. Introduction

The size effects on mechanical properties have been widely investigated in different kinds of materials [1,2,3]. For example, the size effects in the strength of crystalline metals have been investigated to a large extent; and the well-known Hall–Petch relationship has been reported [3]. In contrast to crystalline metals, size effects on the deformation and failure mechanism of amorphous metals have received less attention. Metallic glasses (MGs) or amorphous metals are novel engineering alloys in which the structure is not crystalline, but rather is disordered [4,5,6,7]. Since the first discovery in 1960 [8], MGs have attracted a lot of interest because of their potential as a new type of structural material for practical applications [4,9,10]. However, the lack of macroscopic ductility at room temperature significantly limits their commercial proliferation [11,12,13]. As a consequence, considerable efforts have been made to enhance their deformability [14,15,16,17,18]. One of the most promising strategies is to use feature size as a design parameter, which requires gaining a full understanding of sample size effects on mechanical properties of MGs. For example, a transition from a strong-yet-brittle to a stronger-and-ductile state by the size reduction of MGs to hundreds of nanometers has been reported [19,20,21]. Furthermore, understanding the effects of MG sample size in the micrometer to nanometer regime on mechanical response can provide not only insight into the spatiotemporal evolution of shear transformation zones (STZs), but also guidelines in designing small-volume MGs for microelectromechanical applications [22].

The size effects on the strength and failure mechanism of MGs at different length scales are important to identify and understand. Aside from the brittle-to-ductile transition in failure mode through sample size changes from a micron-to-nano size regime, many experimental studies on MGs at the micrometer regime have been carried out; moreover, the correlation between sample size and mechanical properties, such as maximum plastic strain, yield strength, and deformation mode, are reported [20,23,24,25,26,27,28,29]. However, most of these experimental reports are inconsistent and controversial. For example, Schuster et al. [23,24] and Lai et al. [25] reported no change in failure mode; but an increase in the yield strength of the micron-sized MGs when compared with their bulk counterparts. They attributed this increase to the fact that a smaller sample has a less defective population. In contrast, Volkert et al. reported a decrease in yield strength as the pillar diameter decreases from 8 micrometers to 140 nanometers [26]. Dubach et al. reported that the yield strength of MG micropillars remains unchanged with varying pillar size or free volume content [27]. Chen et al. also reported that the yield strengths of both MGs micropillars and nanopillars are close to those of their bulk counterparts; in addition, they are essentially size-independent [28]. Contradictory conclusions have been drawn by different research groups, based on various experiments on micron-sized MGs. Thus far, the reason for this conflict is still unclear. These apparently contradictory findings, especially the inverse sample size effect in micron-sized MGs, demand further careful investigation.

Arguably, the above-mentioned conflict in size effect may be caused by intrinsic effects; these arise from microstructural heterogeneities, such as sample processing conditions, cooling rates, composition fluctuations, and especially, flaw populations [30,31,32]. In addition, it may also be associated with extrinsic effects, such as test sample geometries and conditions; i.e., sample size, strain rates, and temperature [31,33]. Ruling out these intrinsic and extrinsic factors that can change the atomic structures, we aim to develop a theoretical framework for further insight into the inherent effect of sample size on the mechanical behaviors of micron-sized MGs. Accordingly, a theoretical model based on the shear banding process is proposed to account for the inverse size effect. The anomalous size effect is suggested to be induced by the failure mode transition in micron-sized MGs, where failure occurs in an immature shear banding process associated with shear bands (SBs) formed only at the surface of micropillar. The size effects on the yield strength and failure mode of MGs at different length scales are summarized.

## 2. Theoretical Scheme for the Size Effect at Different Length Scales

Three different size regimes with different governing deformation mechanisms in MGs are illustrated in Figure 1. For MGs with a diameter *d* > ~10 μm, the inevitable presence of weak flaws introduced during the fabrication process and the inherent structural defects (i.e., liquid-like regions), may easily induce SB initiation [13]. In this case, it is well-known that the deformation mechanism is the catastrophic shear banding with the SBs traversing the entire sample; referred to as the ‘full SB’ hereafter. For micron-sized MGs (0.1 μm < *d* < 10 μm), due to the lower probability of having weak flaws in small samples and the reduced structural defects in size and population [21,25,28], we therefore suggest that the SBs only form and propagate at the micro-pillar surface; referred to as the ‘partial SB’ hereafter. For nanoscale MGs with *d* < ~100 nm, STZs have little chance for percolation into SBs due to the insufficient space in the nanosized sample [13,19,20]. Jang et al. demonstrated that when the sample was smaller than the critical size (~100 nm), homogeneous deformation would prevail more than shear banding [20]. The homogeneous deformation through uniform activations of STZs would dominate the deformation mechanism of nanoscale MGs.

## 3. Results and Discussion

### 3.1. Submillimeter-Sized MGs

The initial elastic deformation is followed by a jerky deformation with the presence of displacement bursts [21,34]. In the experiment, the SBs propagate rapidly and the punch gets left behind; resulting in the distinct displacement burst. Noticeably, it is assumed that the failure of MGs is dominated by a single SB and the strain energy relief associated with the SB propagation is totally used to drive the propagation of the dominant SB. Therefore, the yield strength is determined whether the strain energy relief during shear banding is larger than the energy increment of the SB. The reduced elastic energy ΔU after the onset of the first displacement burst can be obtained by [21]
(1)$$\Delta U=\frac{\sigma_{0}^{2}-\sigma_{1}^{2}}{2E}V$$
where *E* is the Young’s modulus; V is the volume of pillar calculated by $V=\pi \alpha d^{3}/4$; α is the aspect ratio of the sample; and $\sigma_{0}$ and $\sigma_{1}$ are the stresses before and after the first displacement burst, respectively.

It is known that the shear banding process in MGs occurs in stages, consisting of SB formation and SB propagation [13]. Therefore, the energy dissipation during the shear banding process ($U_{FSB}$) can be calculated by
(2)$$U_{FSB}=U_{F}+U_{S}$$
where $U_{F}$ and $U_{S}$ represent the energies for SB formation and propagation, respectively. By analogy with crack formation (Griffith criterion), $U_{F}$ has been obtained as [26]
(3)$$U_{F}=\frac{\sqrt{2}\pi d^{2}\Gamma }{4}$$
where Γ is the SB energy density per unit area.

Assuming that the stress $\sigma_{1}$ maintains the SB propagation, $U_{S}$ can be calculated as
(4)$$U_{S}=\frac{\sigma_{1}\pi d^{2}\Delta x}{4}$$
where Δx is the size of the shear offset. Cheng et al. have found that the shear offset size is proportional to the sample length [34]. Since the aspect ratio of MG micro-pillars is fixed, the shear offset size Δx can be expressed as Δx=μd, where μ is the scale factor.

Combining Equations (1)–(4), the yield strength of submillimeter-sized MGs $\sigma_{y}$, which is the stress before the first displacement burst, can be obtained as
(5)$$\sigma_{y}=\sqrt{\sigma_{1}^{2}+\frac{2E}{\alpha }\left(\frac{\sqrt{2}\Gamma }{d}+\mu \sigma_{1}\right)}$$

Taking PdSi MGs for an example, *E*, α, and $\sigma_{1}$ extracted from the work of Volkert et al. are 66 GPa, 2.5, and 1.7 GPa, respectively [26]. The value for the SB energy density per unit area Γ is estimated to be 10 J/m^2^ [26]. The scale factor μ between the shear offset size and sample diameter is obtained as ~0.03, which is consistent with the experimental value of Liu et al. [35]. Figure 2 plots the yield strength as a function of sample size. It is shown that the yield strength has a very weak sample-size dependence in submillimeter-sized MGs, which is consistent with previous work [27].

### 3.2. Micron-Sized MGs

Volkert et al. have reported that the yield strength decreases monotonically when the pillar diameter decreases from 8 μm to 0.14 μm [26], which is obviously different from some experimental reports [20,23,24]. Unfortunately, they did not provide an explanation. They postulated that this could be attributed to the differences in pillar geometry and the influence of rounding at the top of the pillars [26]. Herein, we propose that the physical origin of this seemingly paradoxical inverse size effect is, in fact, caused by the failure mechanism transition to partial or immature shear banding at the pillar surface; leading to a decrease in yield strength. Considering that the SBs do not traverse the entire micropillar and only exist at the surface, the energy dissipation during the formation and propagation of partial SB ($U_{PSB}$) can be obtained by
(6)$$U_{PSB}=\beta \left(U_{F}+U_{S}\right)$$
where β denotes the percentage of lateral thickness of SB in the cross section. β=1 means that the SB traverses the entire micro-pillar, while β=0 means that shear banding gives way to homogeneous deformation. β is proposed as [36]
(7)$$\beta =\frac{d-d_{H}}{d_{FSB}-d_{H}}$$
where $d_{FSB}$ and $d_{H}$ denote the critical sample sizes for the failure modes of full SB and homogeneous deformation, respectively. The critical threshold values of $d_{FSB}$ and $d_{H}$ have been reported as ~10 μm and ~0.1 μm, respectively [21]. Combining Equations (1)–(4), (6), and (7), one can obtain the yield strength $\sigma_{y}$ in micron-sized MGs as
(8)$$\sigma_{y}=\sqrt{\sigma_{1}^{2}+\frac{2\beta E}{\alpha }\left(\frac{\sqrt{2}\Gamma }{d}+\mu \sigma_{1}\right)}$$

In order to verify our theoretical prediction, the yield strengths extracted from the experimental results of Volkert et al. [26] and Dubach et al. [27] are plotted in Figure 3. *E*, α, and $\sigma_{1}$ of ZrTiCuNiBe MGs extracted from the work of Dubach et al. are 92 GPa, 2.5, and 1.9 GPa, respectively [27]. The theoretical prediction shows the same trend as that from experimental results. The decrease in yield strength with decreasing sample diameter could be attributed to the notable change in the deformation mechanism. The partial SB may reduce the energy dissipation during the shear banding process and result in the reduction of yield strength.

In order to justify our model assumptions, it is instructive to discuss several extrinsic factors; these include the tapering angle, geometry of pillars, and surface strain on the mechanical behaviors of micron-sized MGs. First of all, the appearance of a tapering angle is unavoidable for the focus ion beam (FIB) milling procedure; which has been proven to have a significant influence on the failure and deformation behavior of MG pillars under compression testing [24,28]. The induced stress gradient may constrain the nucleation and subsequent propagation of SB; leading to the multiple SBs formation [28,37]. Another extrinsic factor that should be paid more attention is the geometry of the pillars, namely tapering. Due to the tapering effects, SBs are often observed at the corner of the sample–anvil interface [25,38]. In addition to the tapering effects, the surface strain caused by FIB damage is unavoidable; in turn, this facilitates the formation of constrained SBs [39]. These constrained SBs are usually small and stable; as such, they are less likely to run through the sample. In our model assumption on the partial or immature shear banding, the propagated SBs only form at the surface of micropillar.

Furthermore, a core-shell model illustrated in Figure 4 can be used to verify our theoretical model; because the FIB technique can cause sample surface damage due to the Ga ion penetration [25,39], resulting in relatively soft sample surfaces [39]. Two steps are needed for the preparation of micron-sized samples by FIB techniques: milling the sample to the approximate shape with relatively higher beam currents, and removing any taper of the sample sidewalls with finer beam currents [40]. Especially, the sample is tilted in order to mill away the visible taper of the sidewall along one side of the sample in the second step. The depth of Ga ion penetration is fixed in this step. Therefore, the affected region by FIB during sample preparation tends to keep constant. The thickness of surface damage layers is determined by the factors of acceleration voltage, current, and time used in the second step [25]. In view of the results of Bei et al. [41], the FIB damage is non-negligible in the work of Volkert et al. and Dubach et al.; this is because the voltage times current they employed is larger than 30 kV nA [26,27]. As a result, the MG micropillar can be considered to consist of a soft outer shell and hard inner core. It is reasonable to use the core-shell model to investigate the mechanical behaviors of micron-sized MGs. The yield strength of the MG micropillar can be estimated as
(9)$$\sigma_{y}=p_{c}\sigma_{core}+p_{s}\sigma_{shell}$$
where $\sigma_{core}$ and $\sigma_{shell}$ are the yield strengths of the core and shell, respectively. In addition, $p_{c}$ and $p_{s}$ are the volume percentages of the core and shell, respectively. $p_{c}$ and $p_{s}$ are proposed as [42]
(10)$$p_{c}=\frac{{\left(d-2t_{0}\right)}^{2}}{d^{2}}\;p_{s}=1-\frac{{\left(d-2t_{0}\right)}^{2}}{d^{2}}$$
where $t_{0}$ is the thickness of the damaged surface layer. Here, because the factors used in the second step of sample fabrication by FIB remain unchanged, $t_{0}$ is constant; and has been estimated to be ~100 nm from the previous results in literature [25]. $\sigma_{core}$ is taken as ~1.9 GPa, and ~2.1 GPa is based on the yield strength of bulk PdSi MGs [26] and ZrTiCuNiBe MGs [27], respectively. $\sigma_{shell}$ is taken as ~1.4 GPa and ~1.8 GPa extracted from the curve fitting in Figure 4b,c for PdSi and ZrTiCuNiBe MGs, respectively. Note that the basic trend in Figure 4b,c remains essentially unchanged regardless of what parameters are used. The decreasing trend of yield strength obtained from the core-shell model is found to be well-matched with the experimental results, as shown in Figure 4b,c. Therefore, the core-shell model can also give a good interpretation of the observed trend of yield strength in the work of Volkert et al. [26] and Dubach et al. [27]; which is essentially consistent with that from the energetic model.

It is intriguing to discuss the difference between the energetic model and the core-shell model for predicting the size effect on micron-sized MGs. The energetic model is based on the assumption that the released strain energy during shear banding is totally utilized for the propagation of a dominant SB. It is necessary to clarify that the initiation of STZs and the formation of tiny SBs during deformation also require some energy; however, they are not considered for convenience. In comparison, the core-shell model is more reasonable. In the core-shell model, the MG micropillar is considered to consist of a soft shell and hard core; this is consistent with the experimental observation that the sample surface is always damaged during the preparation process [28,40]. Moreover, compared with the energetic model, the yield strengths obtained from the core-shell model match better with the experimental results.

### 3.3. Nanoscale MGs

In this size regime, the uniform activations of STZs act as the dominant deformation mechanism. According to the cooperative shearing model [43], the yield shear stress $\tau_{y}$, which is necessary for the activation of an STZ, is estimated as
(11)$$\tau_{y}=G [0.036-0.016{\left(T/T_{g}\right)}^{2/3}]$$
where *G* is the shear modulus. *T* and *T_g_* are the room temperature and glass transition temperature, respectively. At the nanoscale, the surface stress $\tau_{0}$ should be considered explicitly since the surface-to-volume ratio becomes significant [44]. Generally, due to the difficulties in sample preparation and the operation of instruments in small scales, it is hard to obtain nanosized MG samples with an ideal shape of cylinder. In our theoretical model, the MG nanopillar is idealized as the cylinder for convenience. Thus, the yield strengths under uniaxial tension $\sigma_{yT}$ and compression $\sigma_{yC}$ for nano-sized MGs can be rewritten as
(12)$$\sigma_{yT}=2\tau_{y}+\frac{2\tau_{0}}{d}\;\sigma_{yC}=2\tau_{y}-\frac{2\tau_{0}}{d}$$
with G=23.4 GPa, T=300 K, and $T_{g}=643\;K$ in PdSi MGs [43,45]; $\tau_{y}$ is calculated to be 0.62 GPa. The surface stress $\tau_{0}$ is estimated to be 1.052 J/m^2^ from the work of Zhou et al. [44]. The yield strengths for the nano-sized PdSi MGs under uniaxial tension and compression are plotted as a function of sample diameter in Figure 5. With the decrease of sample size, there is shown to be an increase in $\sigma_{yT}$ and a decrease in $\sigma_{yC}$ for nano-sized PdSi MGs. This finding is consistent with previous MD simulation results on nano-sized CuZr MGs [44]. As discussed by Zhou et al. [44] and Lin et al. [46], the yield strength under compressive loading is reduced compared to that under tensile loading; this is due to the fact that the STZs are easier to develop under compression because of the surface stress exerted on the surface atoms.

From the above discussions, there are three different trends of size-dependent yield strength and the governing deformation mechanisms in MGs. In particular, for the submillimeter-sized MGs, the yield strength is insensitive to sample size and the dominant failure mode is the SB traversing the entire sample. For the micron-sized MGs, the yield strength decreases with sample size, as observed in the work of Volkert et al. [26] and Dubach et al. [27]. The propagated SBs only formed at the MG micropillar surface act as the dominant deformation mechanism, which may dissipate less elastic energy. The reduced energy dissipation during the shear banding process contributed to this anomalous size effect in the micron-sized MGs. For the nanoscale MGs, homogeneous deformation by the uniform activation of STZs plays the dominant role in the deformation mechanism. Consistent with Zhou et al.’s work [44], the surface stress triggers an asymmetry in tensile and compressive yield strengths.

## 4. Conclusions

In summary, there have been considerable efforts to study the size effect in MGs over the past few years due to both scientific interest and technological significance. However, previous experimental studies in micron-sized MGs have reported conflicting results of increased, insensitive, or decreased yield strength with decreasing sample size. A consensus on the correlation between sample size and mechanical properties has not yet been established. Among previous studies, Volkert et al. and Dubach et al. have reported an anomalous size effect in micron-sized MGs; where the yield strength decreases with sample size. Our present study attempts to propose an energetic model and a core-shell configuration model to rationalize this inverse size effect; these are well-matched with experimental results. This anomalous size effect is suggested to be attributed to the change in the deformation mechanism, where the SB only forms at the micropillar surface. Moreover, we summarize the size effects on the yield strength of MGs in three different size regimes; i.e., the size independence regime in submillimeter-sized MGs, the decreasing regime in micron-sized MGs, and the compression/tension asymmetry regime in nanoscale MGs. The associated failure mechanism in these three size regimes is the full SB in submillimeter-sized MGs, the partial SB in micron-sized MGs, and the homogeneous deformation in nanoscale MGs, respectively. The present study contributes to the current understanding of size effects in MGs at different length scales, which is also helpful to interpret the previous experimental observations.

## Figures and Tables

**Figure 1 materials-15-06362-f001:**
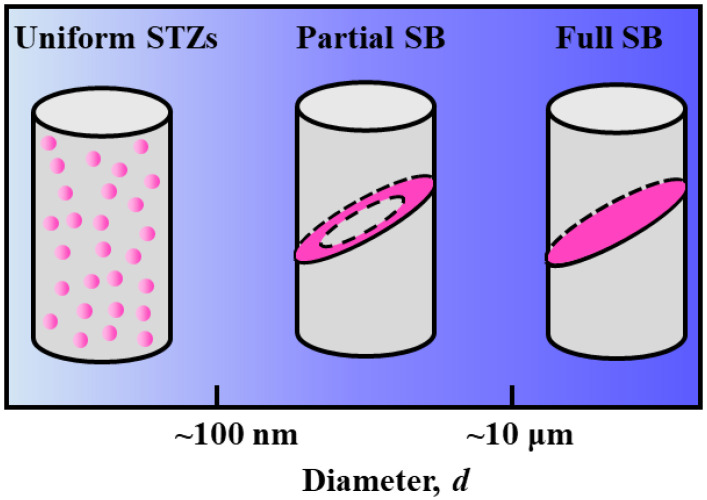
Schematic illustration of the different size regimes and the governing deformation mechanisms in MGs; namely the full SB in submillimeter-sized MGs, the partial SB in micron-sized MGs, and the homogeneous deformation in nanoscale MGs.

**Figure 2 materials-15-06362-f002:**
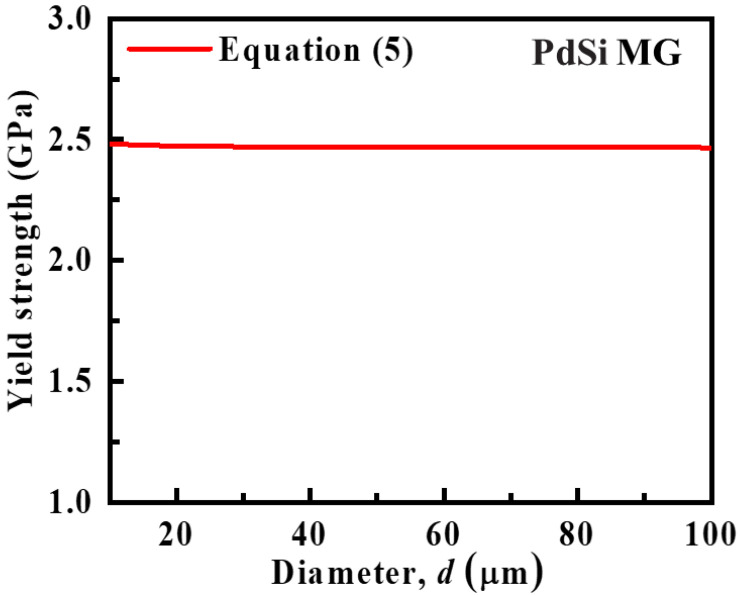
Yield strength as a function of sample size for submillimeter-sized MGs.

**Figure 3 materials-15-06362-f003:**
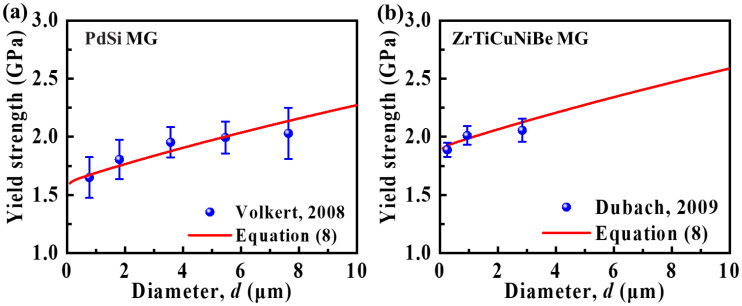
The yield strength as a function of micropillar diameter extracted from the experimental results of (**a**) Volkert et al. [26] and (**b**) Dubach et al. [27]. Our theoretical prediction is also plotted for comparison.

**Figure 4 materials-15-06362-f004:**
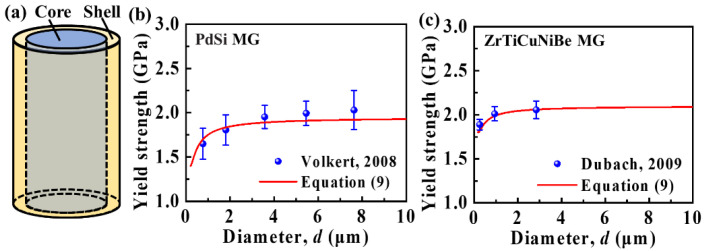
(**a**) Schematic illustration of the core-shell model. The outer shell is the FIB damaged layer, while the inner core is the undamaged MGs. For comparison with the core-shell model, the yield strengths are a function of the sample diameter for micron-sized MGs obtained from the experimental results of (**b**) Volkert et al. [26] and (**c**) Dubach et al. [27].

**Figure 5 materials-15-06362-f005:**
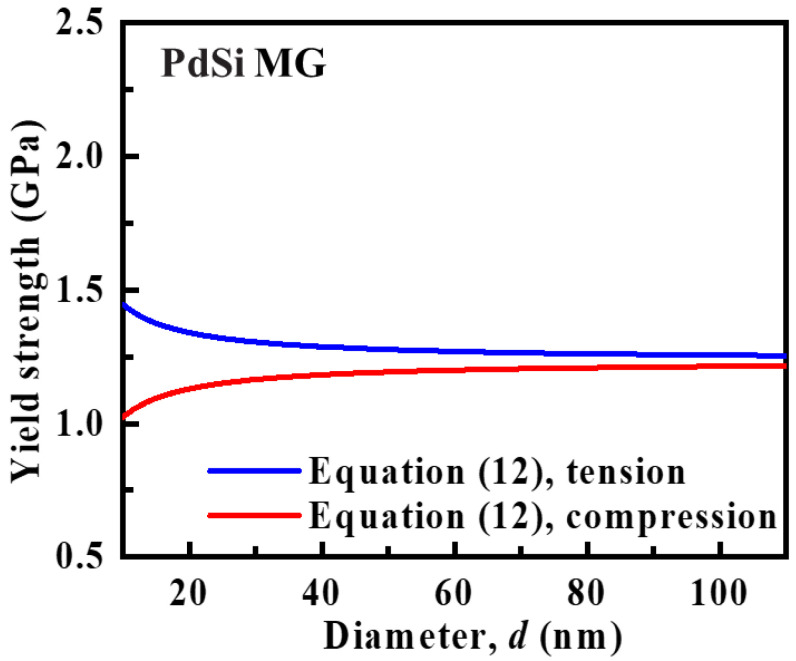
Yield strengths under uniaxial tension and compression as a function of sample diameter for the nano-sized MGs.

## Data Availability

The data used to support the findings of this study are available from the corresponding author upon request.

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
