# Peer review of "Uncovering the Inherent Size Dependence of Yield Strength and Failure Mechanism in Micron-Sized Metallic Glass"

_materials, 2022, doi:10.3390/ma15186362_

Round 1

Reviewer 1 Report

The manuscript is proposing a new theoretical scheme to predict the relationship between yield strength and the size of micron-sized metallic glasses (MGs). The paper is focused on an interesting topic in the field of MGs and deserves publication. However, there are several issues that have to be taken into account before publication.

1- The first point is that is the size-dependent mechanical response in MGs an 'intrinsic' or an 'extrinsic' characteristic?

The substantial variation in properties and performance of materials with different characteristic length scales is known as the size effect. There are two types of size effect: intrinsic and extrinsic. Intrinsic size effects, which have long been the subject of materials studies, arise from microstructural heterogeneities. For example, the strength of pure metals and alloys can be altered over more than two orders of magnitude by introducing soluble alloying elements, second phase precipitates, and grains of varying sizes. In contrast, extrinsic size effects came to materials researchers’ attention more recently. This type of size effect is associated with dimensional constraints, which are of increasing relevance in the age of micro and nanotechnology.
Regarding the assumption of authors that the size-dependency of yield strength in MG is an intrinsic characteristic, some explanation should be added in the Introduction part.

2- While the authors have mentioned the "failure mechanism" in the manuscript title, the main content of the manuscript is only about the size-dependent yield strength in MGs. I think it should be removed from the topic or the authors should add an extensive part to the manuscript to discuss the size-dependent failure mechanism.

3- Section 2 (Theoretical scheme for the size effect at different length scales) is not well organized. The authors should describe how have they reached this point to divide the size-dependency of MG micro-sized samples into three categories. Also, no references have been cited. This section should be thoroughly improved.

4- The title for section 3.1 is "Bulk MGs", and this title is totally incorrect to call micro-sized MGs with diameters in the range shown in Fig. 2. The conventional size to call an MG product as Bulk MG is to have a minimum length/diameter of 1 mm. This title should not be used in this manuscript as the focus of the work is on micro-sized MGs.

5- For equations (7) and (10), the authors should cite related references.

6- Considering Eq. (10), The authors are assuming that the thickness of the surface damage layer (t0) in micro-sized MGs does not depend on the size and is approximated to be ~100 nm. However, this could not be correct. First, the authors have already mentioned that t0 depends on the factors of acceleration voltage, current, and time. Further, assuming that all these factors are constant, as the sample diameter decreases, tmay get larger as the total volume is decreasing and a larger portion of MG diameter could be affected by FIB during sample preparation. So, the authors should give more explanation how they could assume a constant t0 of ~100 nm for all MG of different sizes?

7- It seems that the proposed models in this work only represent a qualitative approach to predicting the size-dependent yield strength trends in MG, and a quantitative evaluation is missing. The authors have only used the experimental data from one article (Ref. 29) to verify their developed models, but this is not enough. More experimental results should be used to verify their proposed model quantitatively.

8- A comparison between the core-shell model and the energetic model should be discussed in Section 3.2 to clarify which one gives a better prediction.

9- For the evaluation of the theoretical approach proposed for nanoscale MGs (Section 3.3), the experimental results should be provided in Fig. 5 (similar to the results given in Fig. 3 and Fig 4(b)).

Author Response

We do appreciate the reviewer's critical comments which have helped improve our manuscript. Following the reviewer's suggestion, we have revised the manuscript.

Reviewer 2 Report

The authors touched upon a very relevant and important topic.

1. The authors in the introduction should describe the sentence in more detail: “As a consequence, considerable efforts have been applied towards enhancing their deformability [14-21].” There are too many references in this sentence.

2. It seemed to me that the methods of analysis and modeling were not sufficiently described.

3. The results are interpreted correctly. The conclusions are substantiated and confirmed by the results

4. Data and analysis provided correctly

5. The methods, tools, software with which this work was carried out are not described in sufficient detail.

Author Response

We appreciate the suggestions by the reviewer. Following the reviewer's suggestion, we have revised the manuscript.

Round 2

Reviewer 1 Report

The authors have well responded to the comments. I recommend acceptance in the present form.